# Active Damping, Vibration Isolation, and Shape Control of Space Structures: A Tutorial

## André Preumont

Department of Control Engineering and System Analysis, Université Libre de Bruxelles (ULB), Avenue F.D. Roosevelt 50, 1050 Brussels, Belgium; andre.preumont@ulb.be

**Abstract:** This tutorial reviews the author's contributions to the active control of precision space structures over the past 35 years. It is based on the *Santini lecture* presented at the IAC-2022 Astronautical Congress in Paris in September 2022. The first part is devoted to the active damping of space trusses with an emphasis on robustness. Guaranteed stability is achieved by using decentralized collocated actuator–sensor pairs. The so-called *integral force feedback* (IFF) is simple, robust, and effective, and the performances can be predicted easily with simple formulae based on modal analyses. These predictions have been confirmed by numerous experiments. The damping strategy for trusses has been extended to cable structures, and also confirmed experimentally. The second part addresses the problem of vibration isolation: isolating a sensitive payload from the vibration induced by the spacecraft (i.e., the unbalanced mass of attitude control reaction wheels and gyros). A six-axis isolator based on a Gough–Stewart platform is discussed; once again, the approach emphasizes robustness. Two different solutions are presented: The first one (active isolation) uses a decentralized controller with collocated pairs of the actuator and force sensor, with IFF control. It is demonstrated that this special implementation of the *skyhook,* unlike the classical one, has guaranteed stability, even if the two substructures it connects are flexible (typical of large space structures). A second approach (passive) discusses an electromagnetic implementation of the *relaxation isolator* where the classical dash-pot of the linear damper is substituted by a Maxwell unit, leading to an asymptotic decay rate of $-40$ dB/decade, similar to the skyhook (although much simpler in terms of electronics). The third part of the lecture summarizes more recent work done on the control of flexible mirrors: (i) flat mirrors for adaptive optics (AO) controlled by an array of piezoelectric ceramic (PZT) actuators and (ii) spherical thin shell polymer reflectors controlled by an array of piezoelectric polymer actuators (PVDF-TrFE) aimed at being deployed in space.

**Keywords:** active damping; piezoelectric actuator; cable structures; vibration isolation; Gough–Stewart platform; skyhook damper; relaxation isolator; adaptive optics; piezopolymer

## 1. Introduction

Increasing the collecting area to feed antennas, spectrographs, and other sensing equipment, and improving the line of sight stability and image resolution are the driving forces in the design of satellites. Up to a certain point, this could be achieved passively [1], by clever design and careful material selection. More recently, the progress in electronics, materials, and a better understanding of structural control made it possible to envision alternative solutions based on active control offering possibilities almost without limits [2].

I was initiated to structural control during a one-year visit at Virginia Tech, in the academic year 1985–1986. The group in which I worked had close links with NASA Langley; the subject was hot because of Ronald Reagan's *Star Wars initiative*. After returning to my home country, I was appointed at ULB, where I attempted to build a small lab of structural control to address the control strategies and also the technologies to implement them (I soon understood that numerical studies that were not supported by experiments were of little value). The recent availability of piezoelectric materials brought also the

problem of modeling multiphysics. This paper summarizes part of the results produced by this lab. For what concerns the vibration control of large space structures, because of the high modal density and the small structural damping (no aerodynamic damping, CFRP structures), it became very clear to me that the issue was (spillover) *stability* [3,4] rather than performance and I directed my work on stability robustness which can be achieved by using actuator–sensor configurations leading to alternating poles and zeros. Similarly, robustness was the main concern in vibration isolation.

The following text is based on the teaching material used for many years in my graduate classes and short courses for industry. It is focused on simple concepts that have been confirmed experimentally, from which the reader can develop his own personal ideas for specific applications. The paper does not include a state of the art as such, but the interested reader will find useful additional readings in the paper(s) quoted in every section. The work on spherical shell reflectors is still ongoing (and prospective).

## 2. Active Damping of a Truss Structure

Most space trusses are made of carbon fibers; they exhibit very small structural damping. This section considers the possibility of integrating one or several active struts to enhance the damping in a robust way (Figure 1). The active strut consists of a linear piezoelectric transducer colinear with a force sensor. The test structure used to demonstrate this technology is shown in Figure 2.

Consider a structure with a single discrete piezoelectric transducer (Figure 3); with the classical notations the transducer is governed by

$$\left\{ \begin{array}{c} Q \\ f \end{array} \right\} = \left[ \begin{array}{cc} C(1-k^2) & nd_{33}K_a \\ -nd_{33}K_a & K_a \end{array} \right] \left\{ \begin{array}{c} V \\ b^T x \end{array} \right\} \tag{1}$$

where $Q$ is the electric charge, $C$ is the capacitance with no external loads ($f = 0$), $k^2$ is the electromechanical coupling factor, $n$ is the number of disks in the stack, $d_{33}$ is the piezoelectric coefficient, $K_a$ is the stiffness of the short-circuited strut ($V = 0$), $V$ is the voltage, and $\Delta = b^T x$ is the relative displacement at the extremities of the transducer. The dynamics of the structure are governed by

$$M\ddot{x} + K^* x = -bf \tag{2}$$

where $K^*$ is the stiffness matrix of the structure without the transducer and $b$ is the influence vector of the transducer in the global coordinate system of the structure. The non-zero components of $b$ are the direction cosines of the active bar. The minus sign on the right-hand side of the equation comes from the fact that the force acting on the structure is opposed to that acting on the transducer. Note that the same vector $b$ appears in both equations because the relative displacement is measured along the direction of the force $f$.

Substituting $f$ from the constitutive equation into the second equation, one finds

$$M\ddot{x} + Kx = bK_a\delta \tag{3}$$

where $K = K^* + bb^T K_a$ is the global stiffness matrix of the structure including the piezoelectric transducer in short-circuited conditions (which contributes to $bb^T K_a$); $\delta = nd_{33}V$ is the *piezoelectric free expansion* of the transducer induced by a voltage $V$; $K_a\delta$ is the equivalent piezoelectric loading: the effect of the piezoelectric transducer on the structure consists of a pair of *self-equilibrating* forces applied axially to the ends of the transducer; as for thermal loads, their magnitude is equal to the product of the stiffness of the transducer (in short-circuited conditions) by the unconstrained piezoelectric expansion; this is known as the *thermal analogy*.

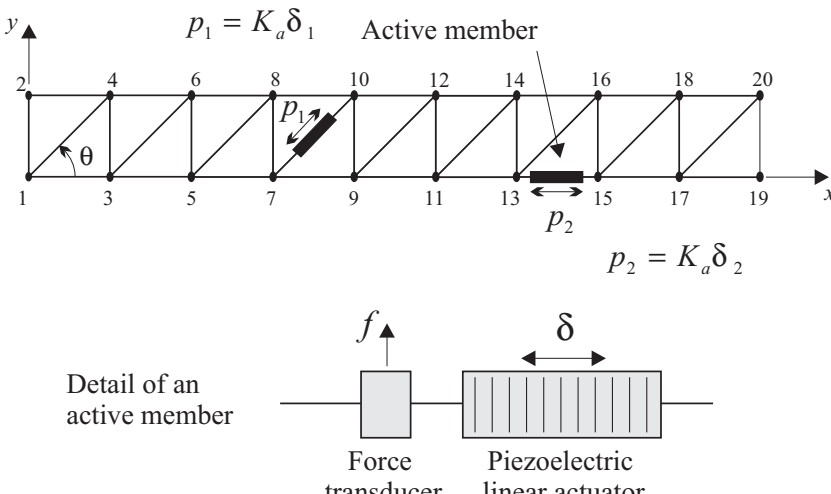

**Figure 1.** Active truss. The active struts consist of a piezoelectric linear actuator with a force sensor.

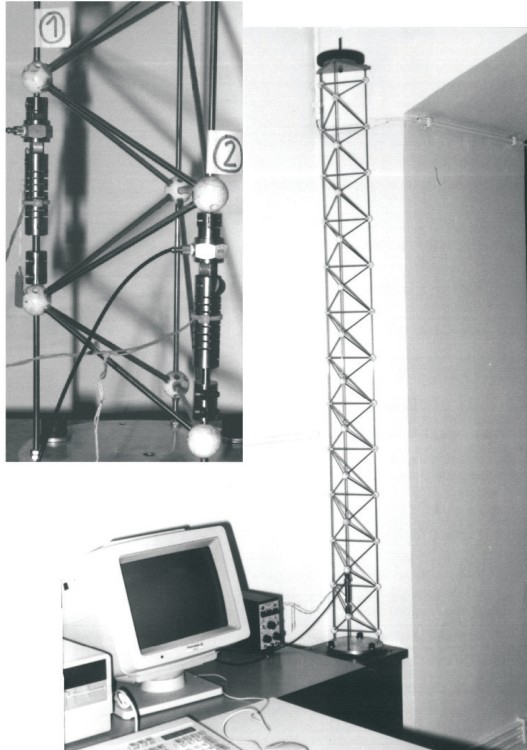

**Figure 2.** ULB Active truss (1988) [5].

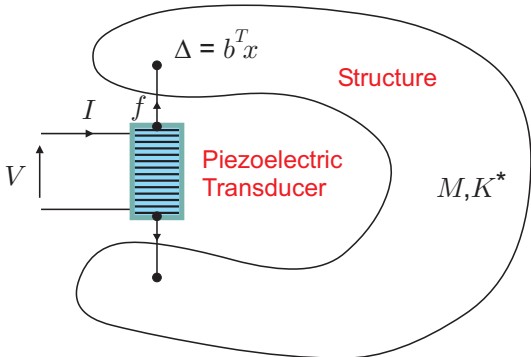

**Figure 3.** Structure with a piezoelectric transducer.

### 2.1. Open Loop

Upon transforming into modal coordinates, one can consider two interesting transfer functions:

The first one is that between the voltage $V$ applied (or the free expansion $\delta$ which is proportional to $V$) and the force $f$ in the strut (it will be used for the force feedback):

$$\frac{f}{\delta} = K_a[\sum_{i=1}^{n} \frac{\nu_i}{(1 + s^2/\omega_i^2)} - 1] \qquad (4)$$

where the sum extends to all of the modes and

$$\nu_i = \frac{K_a(b^T\phi_i)^2}{\mu_i\omega_i^2} \qquad (5)$$

is the *fraction of modal strain energy* in the transducer for mode $i$ ($s$ is the Laplace variable). With all of the residues being positive, there will be alternating poles and zeros along the imaginary axis. Note the presence of a feed-through (constant) in the transfer function. Figure 4a shows the open-loop FRF in the undamped case; as expected the poles at $\pm j\omega_i$ are interlaced with the transmission zeros at $\pm z_i$ ($z_i$ are the natural frequencies of the structure when the active strut has been removed). Equation (4) gives a useful guideline for selecting the most appropriate locations for the active struts: those maximizing $\nu_i$ for the critical modes.

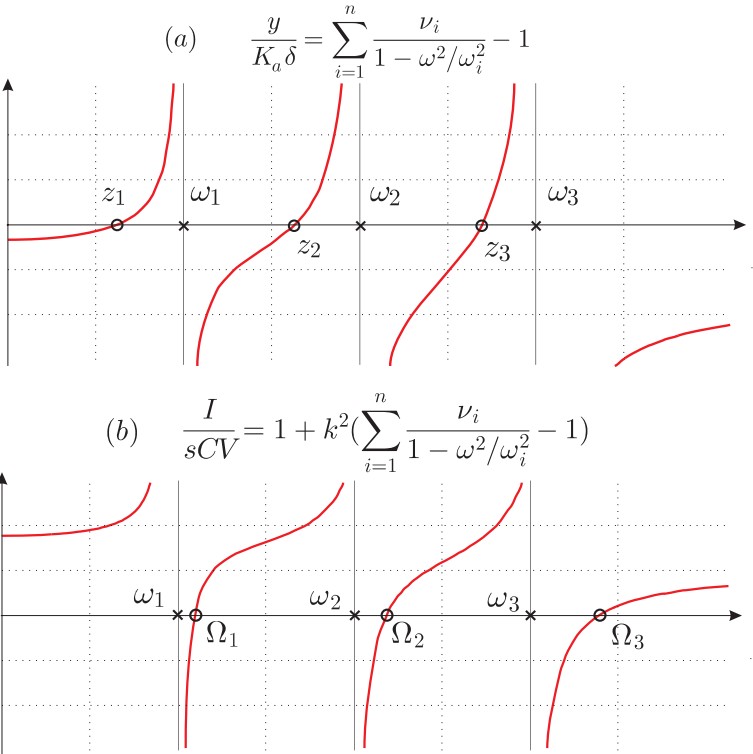

**Figure 4.** (**a**) Open-loop FRF of the active strut mounted in the structure (undamped). (**b**) Admittance of the transducer mounted in the structure.

The second one is that between the voltage $V$ and the current $I$ in the transducer (it is used for damping the truss with shunted $R$ and $L$ elements, or in energy harvesting). Here again, we have alternating poles and zeros (Figure 4b); the poles are the natural frequencies $\omega_i$ of the structure with shunted electrodes ($V = 0$) and the zeros are the natural frequencies $\Omega_i$ of the truss with open electrodes ($I = 0$). All details are available in Section 4.9 of [6].

### 2.2. Closed-Loop: IFF

The key feature associated with an active strut consisting of piezoelectric (displacement) actuator colinear with a force sensor is that the open-loop transfer function between the actuator input (free expansion $\delta$) and the output of the force sensor $f$ has *alternating poles and zeros* (Figure 4a). As a result, the positive integral force feedback (IFF), Figure 5, has *guaranteed stability*.

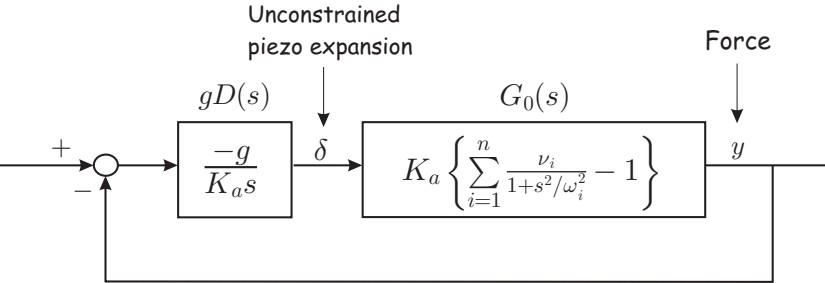

**Figure 5.** Block diagram of the IFF control.

Indeed, Figure 6 shows the corresponding root locus plot showing the evolution of the closed-loop poles when the control gain $g$ increases from zero to $\infty$ (only half of the plot is shown; it is symmetric with respect to the real axis). The closed-loop poles trajectories start from the open-loop poles $\pm j\omega_i$ and end at the transmission zeros $\pm jz_i$. For well separated modes, the individual loops in the root locus of Figure 6 are, to a large extent, independent of each other, and the root locus of a single mode can be drawn from the asymptotic values $\pm j\omega_i$ and $\pm jz_i$ only:

$$1 + g\frac{(s^2 + z_i^2)}{s(s^2 + \omega_i^2)} = 0 \tag{6}$$

It can be shown ([6] pp. 158–159) that the maximum modal damping for mode $i$ is given by

$$\xi_i^{max} = \frac{\omega_i - z_i}{2z_i} \qquad (z_i \geq \omega_i/3) \tag{7}$$

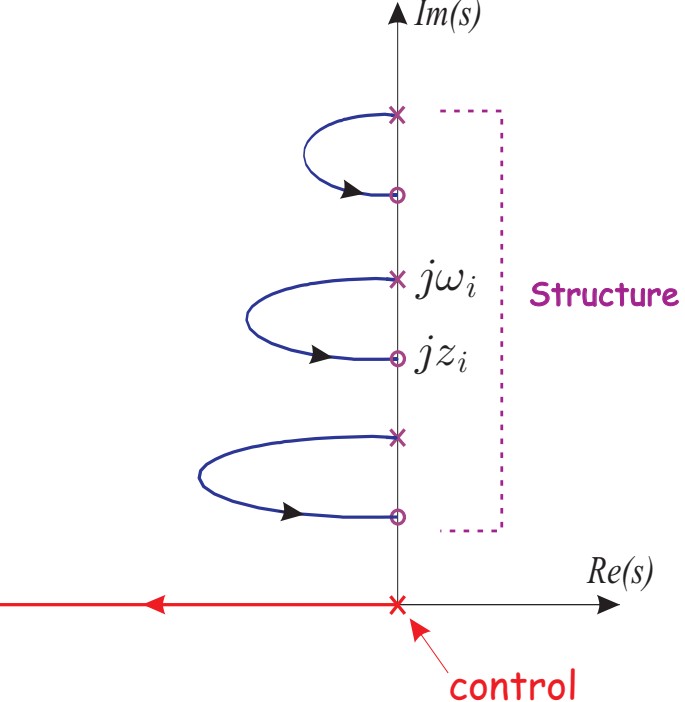

**Figure 6.** Pole-zero pattern of the active strut and root locus of the IFF.

The foregoing discussion assumes perfect actuator and sensor dynamics. The simple form of the IFF allows us to develop straightforward results that have been widely confirmed by experiments. However, a pure integral control is subject to actuator saturation, and the IFF control tends to reduce the static stiffness of the truss. This can be solved by moving the regulator pole slightly on the negative real axis [changing $1/s$ to $1/(s+a)$], or using the *Beta controller* [7]:

$$\frac{\delta}{f} = H(s) = \frac{gs}{K_a(s+\beta)^2} \tag{8}$$

$\beta$ being a small value such that $\beta \ll \omega_1$ (notice that $\beta = 0$ restores the IFF).

The IFF controller was tested successfully on many structures. Figure 7 shows experimental results obtained with the truss in Figure 2. The two active struts have been located in order to target the first two modes of the truss clamped at its base (this can be done simply from the fraction of modal strain energy map $\nu_i$).

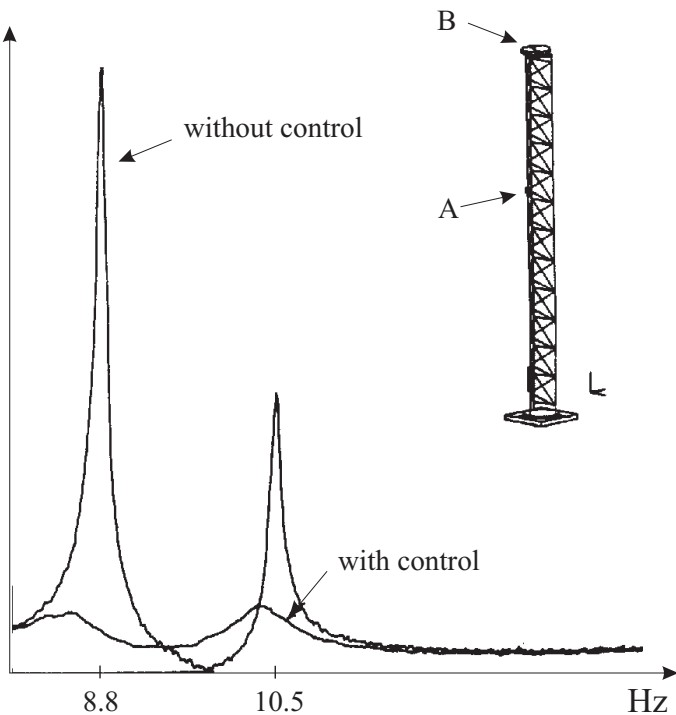

**Figure 7.** FRF between a force disturbance in *A* and the acceleration at *B*, with and without decentralized IFF control (linear scale). The location of the active struts is that of Figure 2 [5].

## 3. Guyed Trusses, Cable Structures

The above theory has been extended to guyed trusses and cable structures (Figure 8) [8–11], including suspension (pedestrian) bridges [12]. It was demonstrated that structures can be damped actively very efficiently with guy cables and, despite the simplifying assumption of the theory, the performances can be predicted with surprising accuracy from the results of two modal analyses, one including all of the cables (Figure 8 left), leading to the open-loop poles at the natural frequencies $\Omega_i$, and one in which all of the active cables have been removed (Figure 8 right), leading to the transmission zeros at the natural frequencies $\omega_i$ (asymptotic values of the closed-loop poles when the control gain goes to infinity).

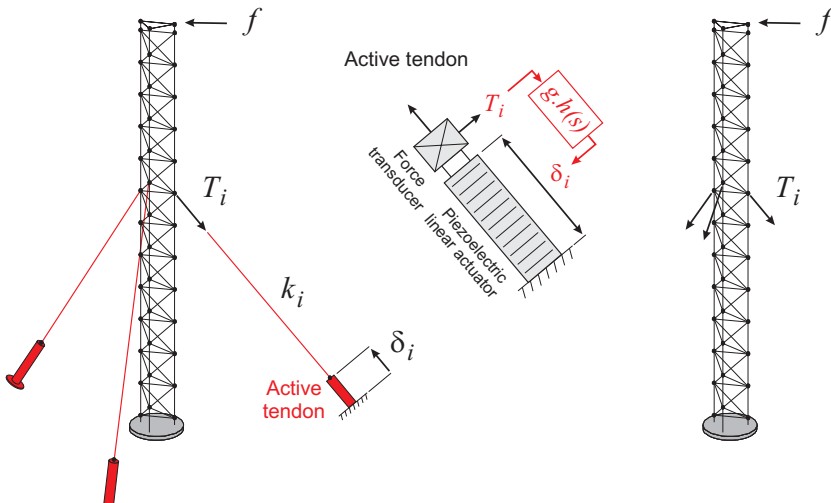

**Figure 8. Left**: Cable structure system with active tendons and decentralized control. **Center**: Active tendon with collocated actuator–sensor. **Right**: Passive structure. $T_i$ is the tension in the active cable $i$ of axial stiffness $k_i$ and free active displacement $\delta_i$.

The maximum achievable damping in mode $i$ is given, once again, by

$$\varsigma_i^{max} = \frac{\Omega_i - \omega_i}{2\omega_i} \tag{9}$$

The previous remark concerning the Beta controller applies fully here too.

### 3.1. NASA–JPL Interferometer Testbed

In order to illustrate how powerful active tendon control can be, consider the micro-precision interferometer (MPI) testbed used at NASA JPL to develop the technology of precision structures for future interferometric missions [13]. The first three flexible modes are displayed in Figure 9 (based on F.E. data kindly provided by JPL). We investigate the possibility of stiffness augmentation and active damping of these modes with a set of three active tendons acting on Kevlar cables of 2 mm diameter, connected as indicated in Figure 10. The global added mass for the three cables is only 110 gr (not including the active tendons and the control system). The natural frequencies of the first three modes, with and without the cables, are reported in Table 1; the root locus of the three global flexible modes as functions of the control gain $g$ are represented in Figure 11; for $g = 116$ rad/s, the modal damping ratios are $\xi_7 = 0.21$, $\xi_8 = 0.16$, $\xi_9 = 0.14$ (the maximum modal damping cannot be achieved simultaneously for all modes, because we assumed that all three control loops have the same gain).

**Table 1.** Natural frequencies (rad/s) of the first flexible modes of the JPL–MPI testbed, with and without cables.

| $i$ | $\omega_i$ | $\Omega_i$ | $\varsigma_i^{max}$ |
|-----|-----------|-----------|---------------------|
| 7 | 51.4 | 74.6 | 0.23 |
| 8 | 76.4 | 101 | 0.16 |
| 9 | 83.3 | 106.4 | 0.14 |

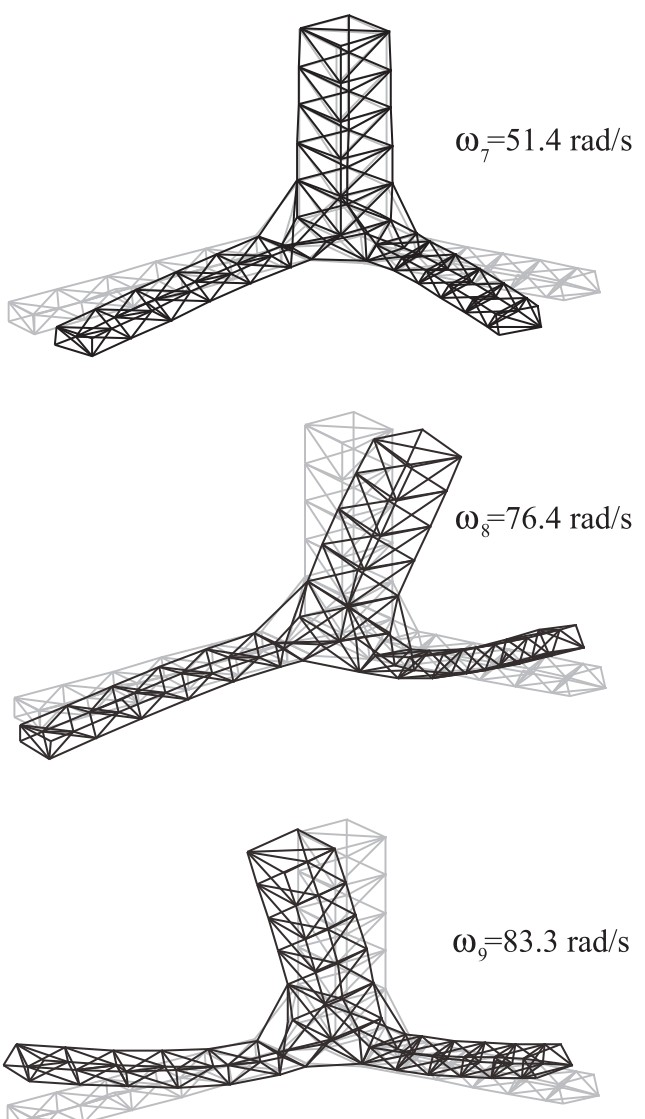

**Figure 9.** JPL–MPI testbed; shape of the first three flexible modes (courtesy of R. Laskin—JPL).

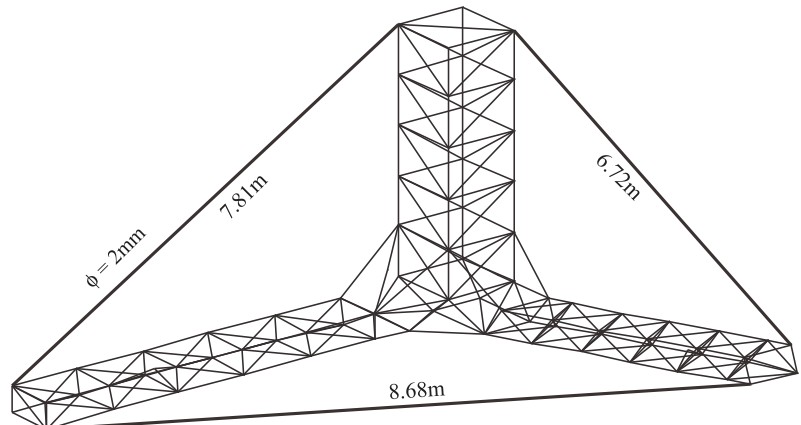

**Figure 10.** Proposed location of the active cables in the JPL–MPI testbed.

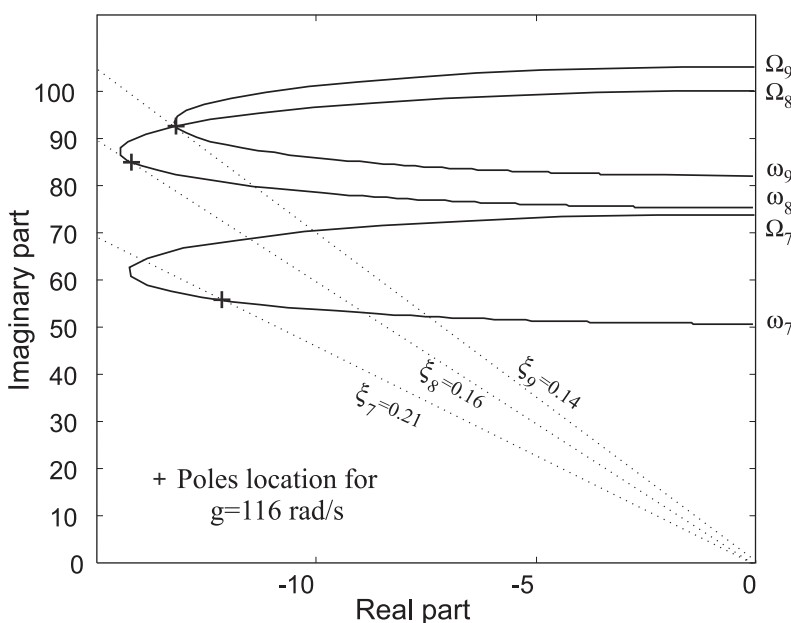

**Figure 11.** Analytical prediction of the closed-loop poles. For $g = 116$ rad/s, the modal damping ratios are $\xi_7 = 0.21$, $\xi_8 = 0.16$, $\xi_9 = 0.14$.

### 3.2. Free-Floating Truss Experiment

In order to confirm the spectacular analytical predictions obtained with the numerical model of the JPL–MPI testbed, a similar structure (although smaller) was built and tested (Figure 12); the free-floating condition was simulated by hanging the structure with soft springs. The active tendon consists of an APA100M amplified actuator from cedrat, together with a B&K 8200 force sensor and flexible tips. The stroke is 110 μm and the total weight of the tendon is 55 gr; the cable is made of Dyneema with axial stiffness $EA = 19,000$ N. The experimental natural frequencies of the first four flexible modes, with and without cables, are reported in Table 2. Every closed-loop pole is located on a loop connecting one pole $\Omega_i$ to one zero $\omega_i$, but we ignore a priori which one; this can be seen only by tracking the poles for increasing values of the gain. Figure 13 compares the analytical predictions of the linear model and the experiments.

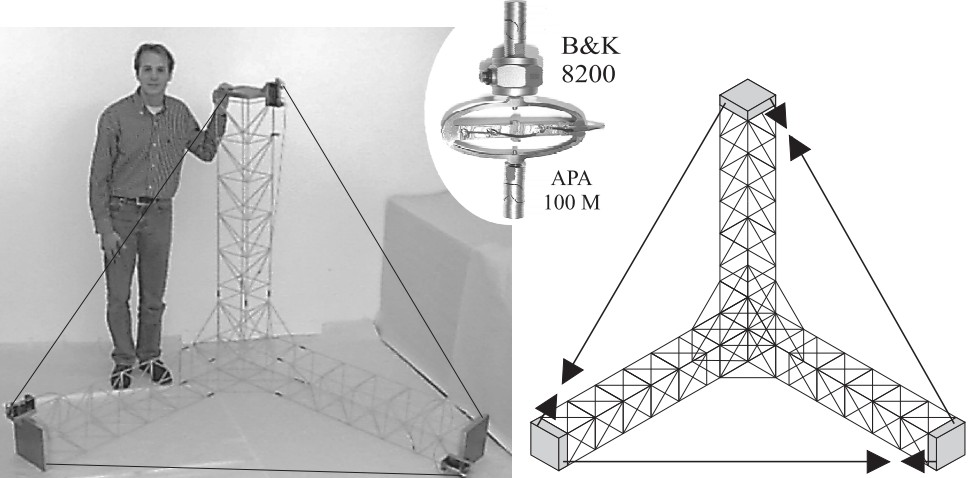

**Figure 12.** ULB free-floating truss test structure and detail of the active tendon [11].

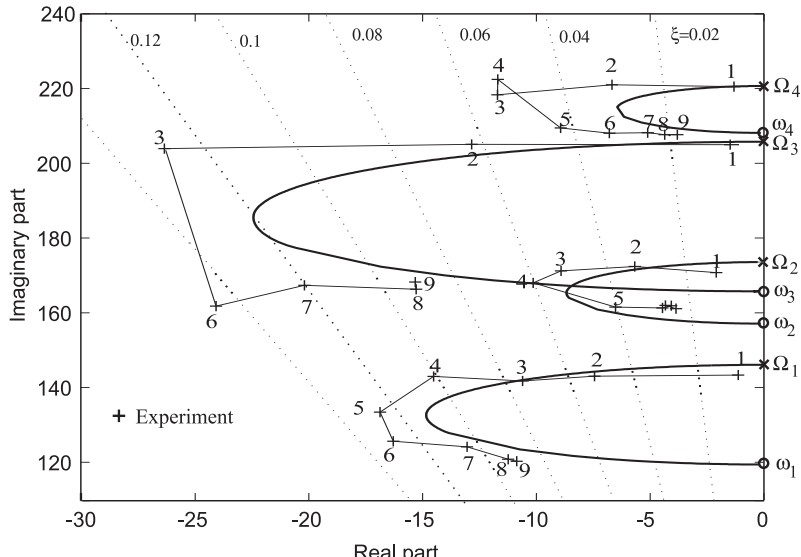

**Figure 13.** ULB free-floating truss test structure: Comparison between the analytical predictions of the linear model and the experiments. The numbers correspond to equal values of the gain.

**Table 2.** ULB truss. Experimental natural frequencies (rad/s) of the free-floating truss, with and without cables.

| $i$ | $\omega_i$ | $\Omega_i$ |
|---|---|---|
| 1 | 119.4 | 146.1 |
| 2 | 157.1 | 173.6 |
| 3 | 165.7 | 205.8 |
| 4 | 208.1 | 220.7 |

Note that all of the results discussed above have been obtained for vibrations in a range from millimeters to microns; in order to apply this technology to future large space platforms for interferometric missions, it is essential that these results be confirmed for vibration amplitudes of only a fraction of the wavelength. In fact, it could well be that, for very small amplitudes, the behavior of the control system is dominated by the nonlinearity of the actuator (hysteresis of the piezo), the noise in the sensor, or in the voltage amplifier. Tests have been conducted for vibrations of decreasing amplitudes, and the influence of the various hardware components has been analyzed [14]; these tests indicate that active damping is feasible at the nanometer level, provided that adequately sensitive components are used.

## 4. Payload Vibration Isolation

Space telescopes and precision payloads are subject to jitter due to the unbalanced masses of the attitude control reaction wheels or gyros. The performance of the instruments may be improved by inserting one or several isolators in the transmission path between the disturbance source and the payload. Referring to Figure 14, the vibration isolator always involves some amplification in the vicinity of its corner frequency, and attenuation at higher frequencies. Classical linear dampers (spring plus viscous damper) offer an attenuation of $-20$ dB/decade ($\sim\omega^{-1}$). In this section, we examine how a high-frequency attenuation of $-40$ dB/decade ($\sim\omega^{-2}$) can be achieved, actively, or passively. An attenuation of $-40$ dB/decade means that, at a frequency one decade larger than the corner frequency $f_0$, the jitter is reduced 100 folds. Extremely sensitive payloads may even involve several isolation layers: The James Webb Space Telescope (JWST) involves two isolation layers, *(i)* the wheel isolator supporting six reaction wheels, with corner frequencies at 7 Hz for

rocking and 12 Hz for translation and *(ii)* a 1 Hz passive isolator at the interface between the telescope deployment tower and the spacecraft bus [15]. In order to fully isolate two rigid bodies, one needs to consider the 6 degrees of freedom. For a number of space applications, it is convenient to integrate the six axes of isolation in a Gough–Stewart platform (Figure 15) in which every leg of the platform consists of a single-axis isolator, connected to the base plates by spherical joints.

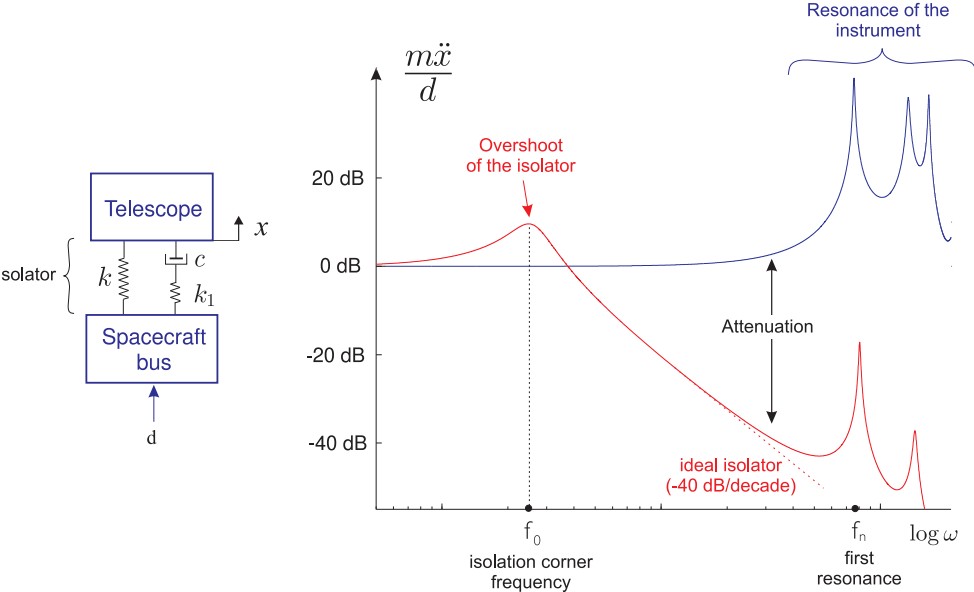

**Figure 14.** Effect of the isolator on the transmissibility between the spacecraft bus and the telescope.

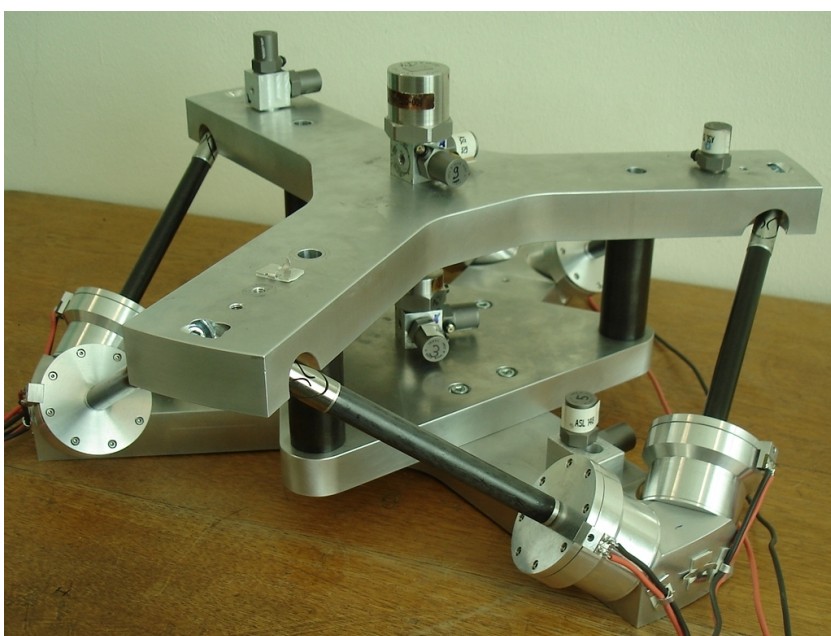

**Figure 15.** Multi-purpose soft (relaxation) isolator based on a Gough–Stewart platform with cubic architecture (ULB) [16].

In the six-axis isolator, we assume that the six legs work independently (decentralized control). We first consider the behavior of a single leg.

## 5. Single-Axis Skyhook Damper

The most popular single-axis active isolator to isolate two rigid bodies is known as the skyhook damper [17] (Figure 16). It consists of a spring *k* in parallel with a force actuator

$F_a$ (e.g., a voice coil) and a precision sensor measuring the absolute acceleration or the absolute velocity (geophone) of the payload. Notice the absence of a viscous damper. The feedback law is such that the control force $F_a$ is proportional to the absolute velocity of the payload (the mass $M$ in Figure 16) [the controller takes its name from the fact that it behaves similar to a viscous damper of constant $g$ attached to a fixed point—the sky]. It is readily established that the transmissibility between the disturbance source and the sensitive equipment has a high-frequency attenuation of $-40$ dB/decade and the overshoot at the corner frequency can be eliminated by a proper selection of the control gain $g$; the root locus of the closed-loop system is represented in Figure 17. The system works very well and is used extensively when dealing with rigid bodies. However, large-space structures rarely behave as rigid bodies because they include very flexible components, such as the solar panels; the skyhook damper does not offer any guarantee of stability when it operates between flexible structures.

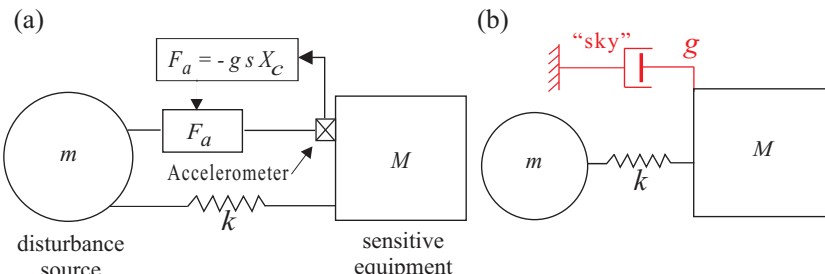

**Figure 16.** (**a**) Isolator based on acceleration or an absolute velocity sensor (geophone). (**b**) Equivalent *skyhook* damper.

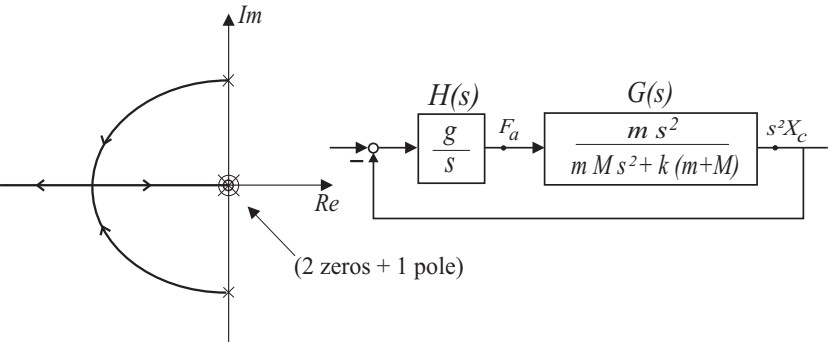

**Figure 17.** Root locus of the *skyhook* damper (acceleration sensor).

*Integral Force Feedback*

As an alternative to the classical skyhook damper of Figure 16, consider the configuration represented in Figure 18, in which the inertial sensor attached to the sensitive equipment has been replaced by a force sensor $F$ measuring the total force applied to the sensitive equipment [18]. We know from Newton's law that, if the two bodies are rigid, the behavior of the isolator will be identical to that of Figure 16. However, the two implementations of the skyhook isolator behave differently if the bodies are flexible (Figure 19): It can be proved that, *if two arbitrary undamped flexible bodies are connected by a single-axis isolator with force feedback and the architecture of Figure 19, the poles and zeros of the open-loop transfer function $F/F_a$ always alternate on the imaginary axis* ([6], pp. 174–175) In other words, the skyhook isolator based on force feedback has guaranteed stability.

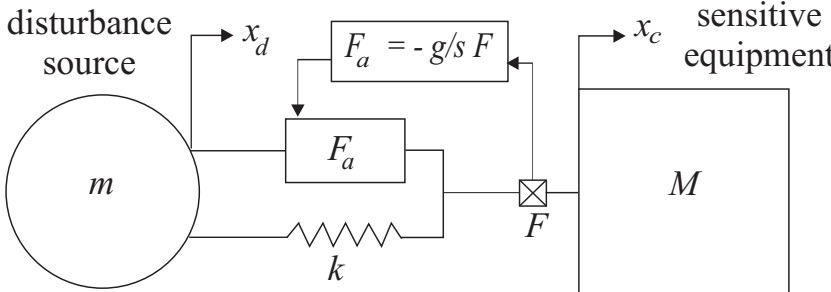

**Figure 18.** Skyhook based on a force sensor ($F$ is positive when it acts in the direction of $x_c$ on mass $M$).

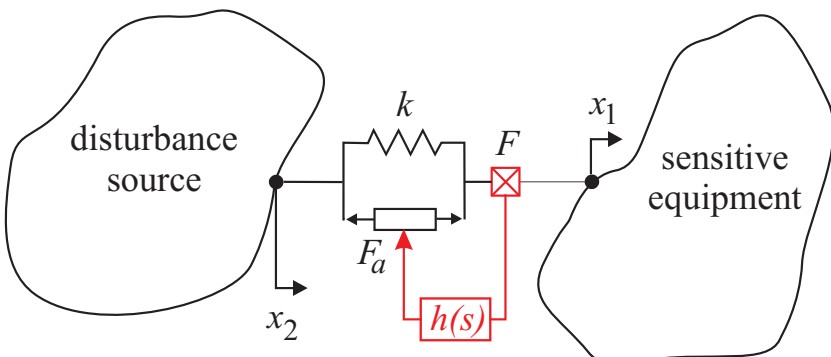

**Figure 19.** Arbitrary flexible structures connected by a single-axis isolator with force feedback. The skyhook corresponds to $h(s) = -g/s$ (integral force feedback).

Note that, besides the advantage of achieving alternating poles and zeros, a force sensor may be more sensitive than an accelerometer in low-frequency applications; for example, a force sensor with a sensitivity of $10^{-3}$ N is commonplace; for a mass $M$ of 1000 kg (e.g., a space telescope), this corresponds to an acceleration of $10^{-6}$ m/s$^2$; such a sensitivity is more difficult to achieve. Force sensing is especially attractive in micro-gravity where one does not have to consider the dead loads of a structure.

## 6. Gough–Stewart Platform

To fully isolate two rigid bodies with respect to each other, six single-axis isolators judiciously placed are needed [19,20]. For a number of space applications, generic multi-purpose isolators have been developed with a standard Gough–Stewart platform architecture, in which every leg of the platform consists of a single-axis active isolator, connected to the base plates by spherical joints. In the cubic architecture (Figure 15), the legs are mutually orthogonal, which minimizes the cross-coupling between them [21]. This configuration is particularly attractive, because it also has uniform stiffness properties and uniform control capability, and it has been adopted in most of the projects.

If one assumes that the isolator consists of six independent legs similar to that of Figure 18 connected with spherical joints and controlled in a decentralized manner with the same gain $g$, it is easy to establish that the six suspension modes follow the characteristic equation

$$1 + g \frac{s}{s^2 + \Omega_i^2} = 0 \tag{10}$$

where $\Omega_i$ are the natural frequencies of the six suspension modes. The corresponding root locus is shown in Figure 20a; it is identical to that of a single-axis isolator; however, unless the 6 natural frequencies are identical, a given value of the gain $g$ will lead to different pole locations for the various modes and it will not be possible to achieve the same damping for all modes. This is why it is recommended to locate the payload in such a way that the spread of the modal frequencies is minimized.

In order to eliminate the backlash associated with the spherical joints, they are replaced by flexible joints with large longitudinal and shear stiffness and low bending stiffness. These flexible joints bring additional contributions to the stiffness matrix with the consequence that the transmission zeros are no longer at the origin but are shifted along the imaginary axes (Figure 20b).

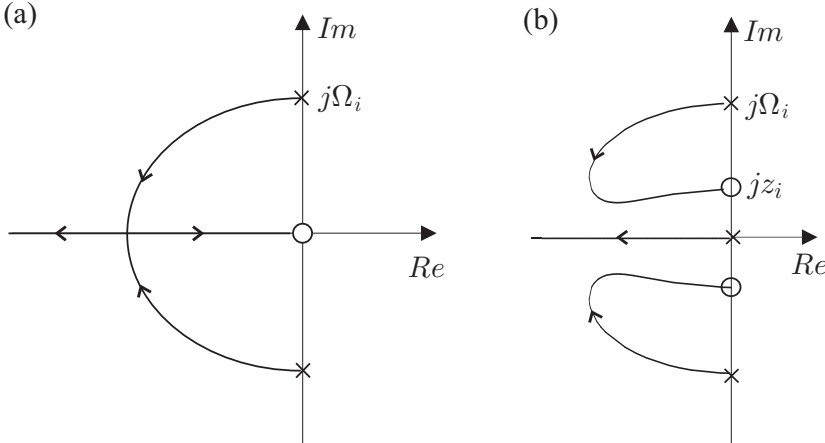

**Figure 20.** Six-axis active isolator with independent IFF loops: root locus of individual modes. (**a**) with perfect spherical joints. (**b**) with flexible joints.

As mentioned before, the six suspension modes have different natural frequencies and the decentralized IFF controller has a single gain $g$ which has to be adjusted to achieve a good compromise in the suspension performance for the six modes. The best performance is achieved if the suspension is designed in such a way that the *modal spread*, $\Omega_6/\Omega_1$, is minimized. The combined effect of the modal spread and the joint stiffness is illustrated in Figure 21; there are only 4 different curves because of the symmetry of the system. The bullets correspond to the closed-loop poles for a fixed value of $g$; they illustrate the fact that the various loops are traveled at different speeds as $g$ increases. How this impacts the transmissibility is examined below.

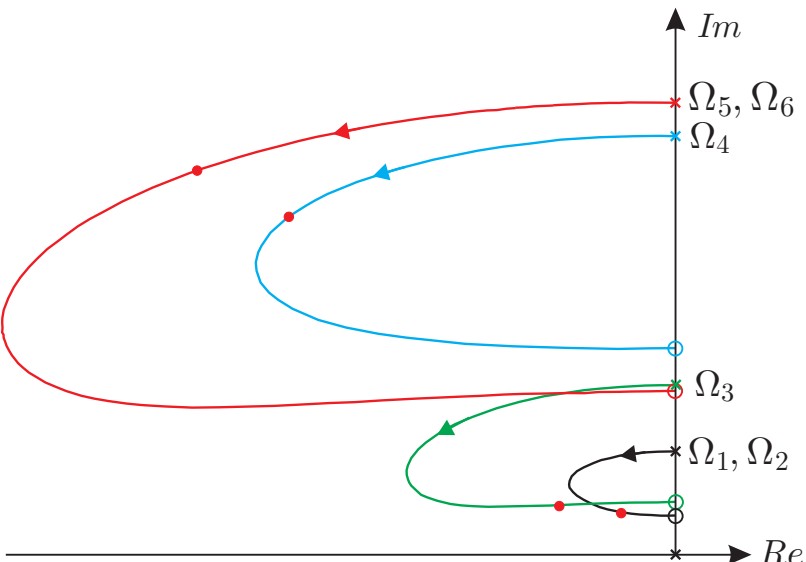

**Figure 21.** Root locus of a complete isolation system with real joints. The bullets indicate the location of the closed-loop poles for the adopted value of the gain $g$.

## 7. Relaxation Isolator

In the *relaxation* isolator, the viscous damper $c$ is replaced by a Maxwell unit consisting of a damper $c$ and a spring $k_1$ in series (Figure 22a).

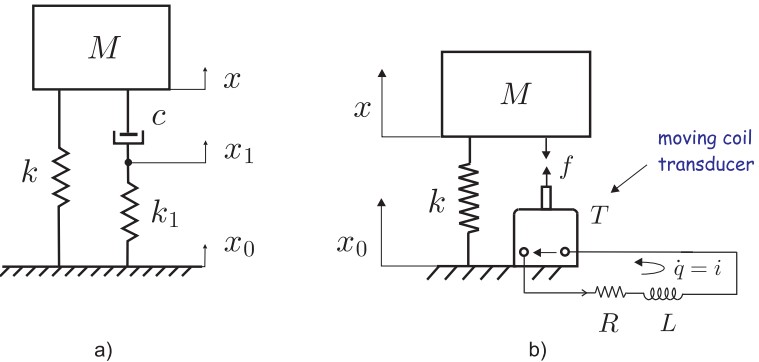

**Figure 22.** (**a**) Relaxation isolator. (**b**) Electromagnetic realization.

For $c = 0$, the relaxation isolator behaves similar to an undamped isolator of natural frequency $\omega_n = (k/M)^{1/2}$. Likewise, for $c \to \infty$, it behaves similar to an undamped isolator of frequency $\Omega_n = [(k + k_1)/M]^{1/2}$. In between, the poles of the system are solutions of the characteristic equation

$$1 + \frac{k_1}{c}\frac{s^2 + \omega_n^2}{s(s^2 + \Omega_n^2)} = 0 \tag{11}$$

the maximum damping ratio ($A$ in Figure 23) is achieved for

$$\frac{k_1}{c} = \frac{\Omega_n^{3/2}}{\omega_n^{1/2}} \tag{12}$$

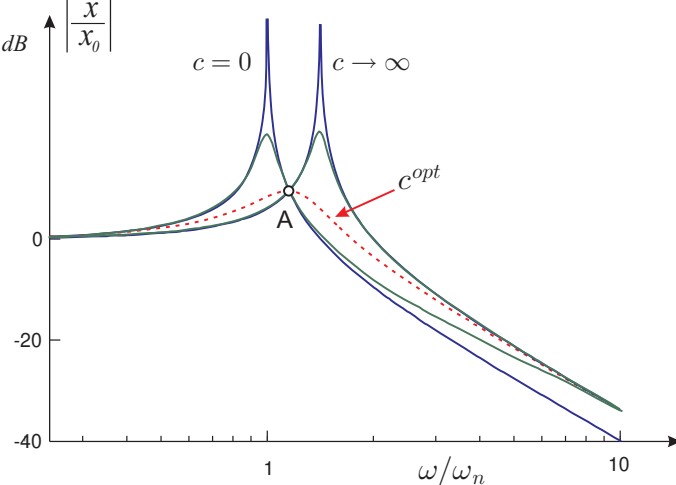

**Figure 23.** Transmissibility of the relaxation isolator for fixed values of $k$ and $k_1$ and various values of $c$. The first peak corresponds to $\omega = \omega_n$; the second one corresponds to $\omega = \Omega_n$. All of the curves cross each other at $A$ and have an asymptotic decay rate of $-40$ dB/decade. The curve corresponding to $c_{opt}$ is nearly maximum at $A$.

The principle of the relaxation isolator is simple and it can be realized with viscoelastic materials. However, it may be difficult to integrate into the system, and also to achieve thermal stability. In some circumstances, especially when thermal stability is critical, it may be more convenient to achieve the same effect through a voice coil transducer of constant $T$,

an inductor $L$ and a resistor $R$ (Figure 22b). The electromechanical isolator behaves exactly as a relaxation isolator provided that [16]

$$k_1 = \frac{T^2}{L} \qquad c = \frac{T^2}{R} \tag{13}$$

Figure 24 compares the components involved in the passive relaxation isolator and the active one. The active isolation requires conditioning electronics for the force sensor and power electronics for the voice coil actuator. The relaxation isolator requires only a passive $RL$ circuit but also requires a bigger transducer (with a larger transducer constant $T$). Moreover, it does not have a force sensor, which makes it lighter. In fact, the legs have their own local dynamics that interfere with that of the isolator and impact significantly the transmissibility in the vicinity of the resonance frequency of the local modes and beyond. Maximizing the natural frequency of the local modes of the legs is a major challenge in the design of a six-axis isolator with broadband isolation capability. This is achieved through careful design of all of the components of the isolator. Figure 25 shows the leg of a passive relaxation isolator; the exploded view of the transducer shows the membrane which acts as an axial spring and also plays the role of the spherical joint, the magnetic circuit, and the voice coil, and its connection to the stinger, made of CFRP to minimize its weight.

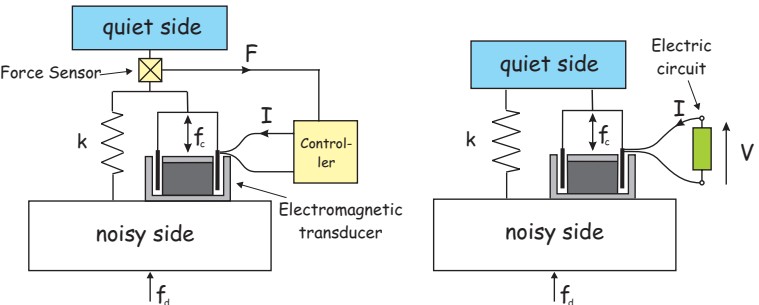

**Figure 24.** Comparison of the active isolator (**left**) with the passive isolator (**right**); if a $RL$ electrical circuit is used, the passive isolator is a relaxation isolator; a purely resistive circuit produces a linear viscous isolator.

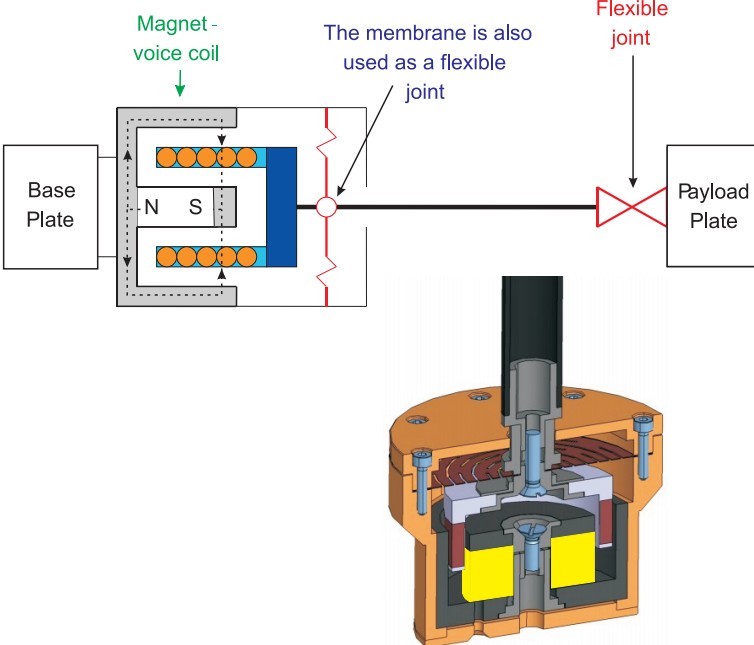

**Figure 25.** Leg of a passive relaxation isolator; conceptual design and exploded view of the transducer showing the membrane, the magnetic circuit, the voice coil, and its connection with the stinger.

We conducted extensive testing on six-axis isolators, including in parabolic flight [19], to simulate the 0-gravity environment. Figure 26 compares the transmissibility of the active (IFF) and the passive (relaxation) isolator. The dotted line refers to the transmissibility of the passive isolator when the *RL* circuit is open. As expected, the overshoot of the active one is a little lower; both have a decay rate of −40 dB/decade in the intermediate frequency range, and the high-frequency behavior is dominated by the local modes; the passive isolator behaves better in high frequency, because the local modes have higher resonance frequencies.

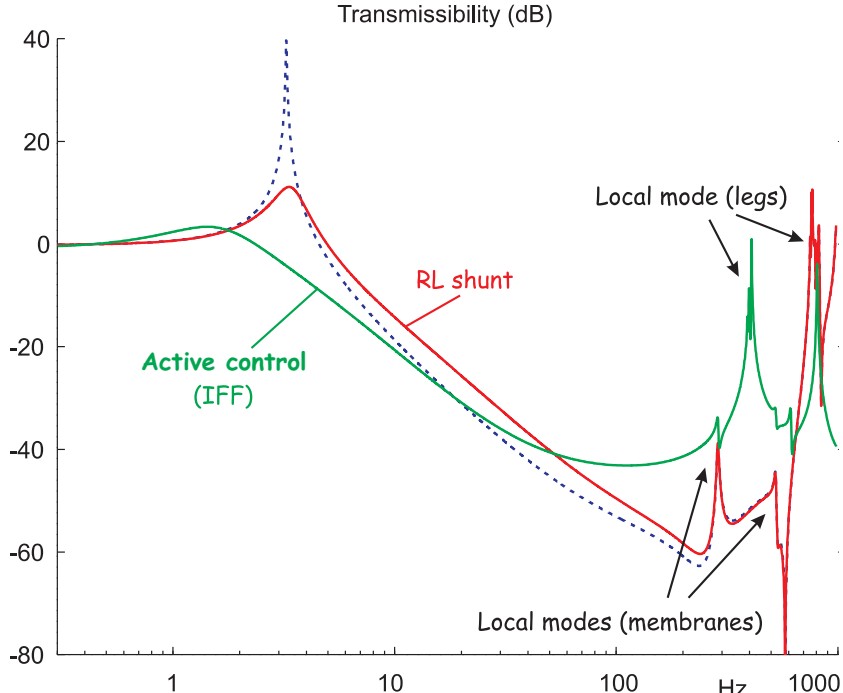

**Figure 26.** Vertical transmissibility of a six-axis isolator: comparison of the Open-loop (dotted line), closed-loop (IFF) active isolator, and passive relaxation isolator with a *RL* shunt (from [16]).

## 8. Shape Control of Precision Structures

Once in orbit, the shape of antenna reflectors and telescopes must be maintained within a fraction of the wavelength, in spite of the manufacturing and deployment errors, gravity release, gravity gradient, and thermal loads (the thermoelastic loads due to thermal gradients are often the main contributors to the shape distortion, especially in LEO, because of the frequent eclipses that induce thermal loads). Maintaining the surface figure accuracy of a deployed reflector is becoming increasingly difficult as the wavelength $\lambda$ becomes shorter (RMS wavefront accuracy of $\lambda/14$ for optical systems) and active wavefront control offers an elegant solution. Possible applications can be found in large reflectors (the challenge is to compensate for the error resulting from unfolding the reflector in addition to the thermal deformations) as well as in small and nano satellites where using deployable light collectors allows for drastically improving the performance of Earth observation, inter-satellite laser communication, or LiDAR. Active wavefront control can be done with adaptive optics (AO) or by direct shape control of the primary reflector.

## 9. Adaptive Optics

Adaptive Optics was invented to compensate for atmospheric turbulence on Earth-based telescopes (i.e., the wavefront distortion due to the random variation of the refraction index). AO uses a flat deformable mirror capable to generate complex shapes of small amplitudes (a few microns). For Earth applications, the control system must be fast, with a bandwidth of 50–100 Hz, depending on the wavelength, the telescope site, the diameter of the primary mirror, and the wind velocity. The same technology may be used to correct the

aberrations introduced by the imperfect shape of the primary mirror (deployment error, thermal deformations, . . . ) although a fast response is not needed in space, the disturbance being slow.

*Piezoelectric Unimorph Mirror*

There exist AO mirrors of various sizes and they can be actuated with various technologies. We focused our attention on a mirror controlled with an array of PZT actuators. A linear relationship is assumed between the voltages $v$ applied to the piezoelectric actuators and the output $s$ of a wavefront sensor (e.g., Shack–Hartmann)

$$s = J v \tag{14}$$

$J$ is a constant rectangular matrix (Jacobian); the sensor vector $s$ is usually larger than the actuator vector $v$. The actuator which minimizes the mean-squares sensor output is given by the pseudo-inverse

$$v = J^{\dagger} s \tag{15}$$

If the Jacobian is ill-conditioned, a Tikhonov regularization may be used in order to reduce the amplitudes of the voltages (see Ch.16 of [6]). Figure 27 shows the most common electrode layouts of unimorph piezoelectric AO mirrors. Figure 28 shows the experimental results of typical aberrations generated with a honeycomb layout and the corresponding voltage maps [22].

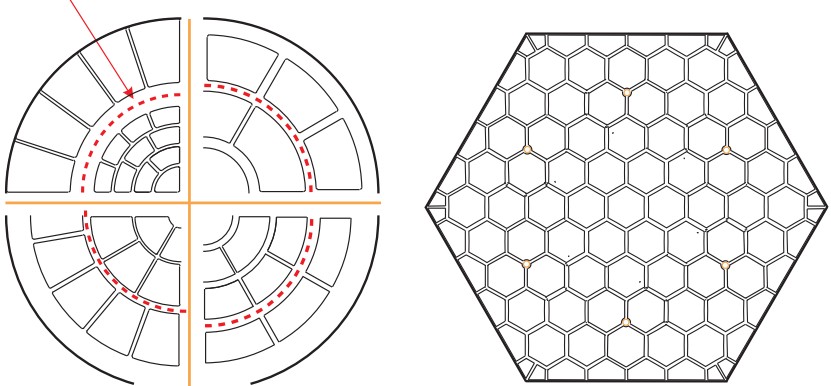

**Figure 27.** Most common electrode layouts of unimorph piezoelectric mirrors. **Left**: the *Keystone* layout is well-suited to controlling the low-order Zernike modes. **Right**: the honeycomb layout is homogeneous and well-suited to scaling up and segmented design.

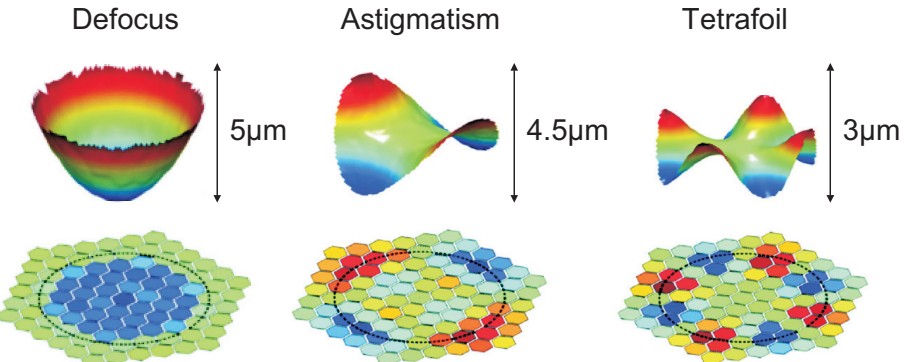

**Figure 28.** Deformable mirror: typical corrected aberrations with the corresponding voltage distributions within the honeycomb electrodes [22].

Figure 29 shows a prototype of an AO mirror actuated with 25 electrodes arranged in a keystone configuration [23]. The mirror consists of a 500 μm thick Silicon wafer of 76.2 mm diameter; the 200 μm PZT patch has a diameter of 50 mm and the optical pupil is 30 mm. The mirror is mounted via isostatic support with piston–tip–tilt actuators (APA50XS from cedrat). The 25 electrodes are obtained by laser ablation. The 25 electrodes appear sufficient to control 15 Zernike modes [24]. Figure 30 shows a comparison between the experimental results and a simulation for a 1 μm astigmatism.

Using unimorph PZT mirrors brings two technological problems: The first one stems from the fact that PZT actuators use only positive voltage (to avoid depolarization). This requires the application of a constant bias voltage. The problem is solved by applying the bias voltage during the gluing process of the PZT patch at the back of the mirror (the mirror will be slightly concave after gluing, but it will recover its flat shape when the bias voltage is applied).

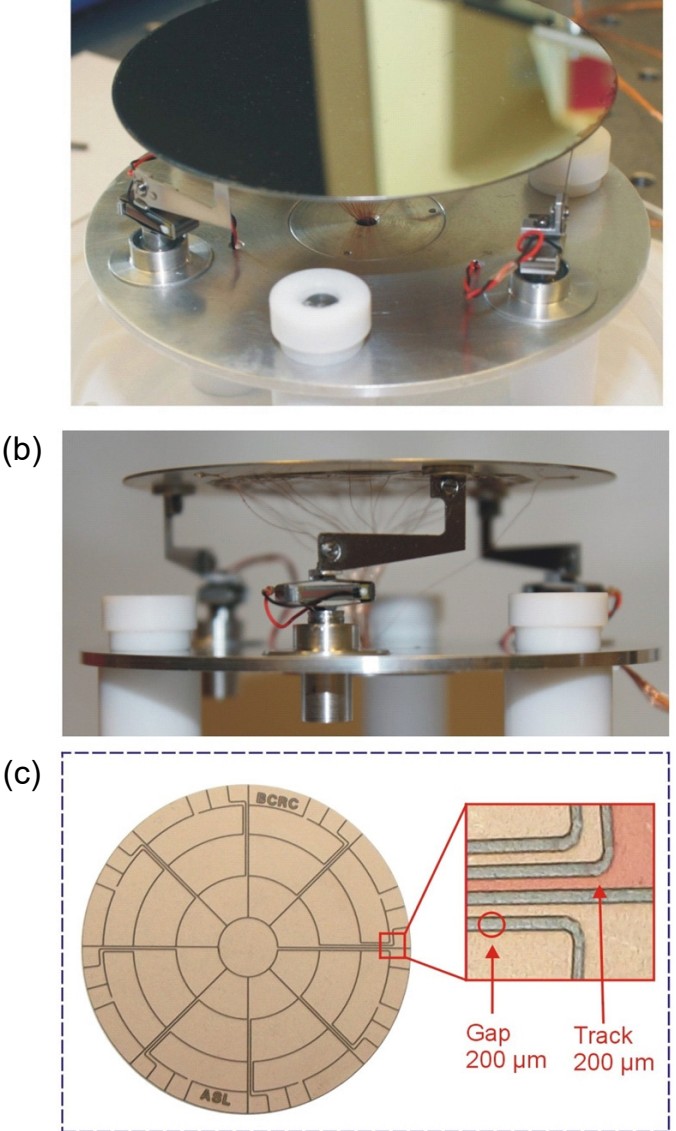

**Figure 29.** (**a**) Prototype AO mirror actuated with 25 electrodes in keystone configuration. (**b**) Detail of the isostatic mount with APA50XS actuators. (**c**) Keystone electrodes obtained by laser ablation.

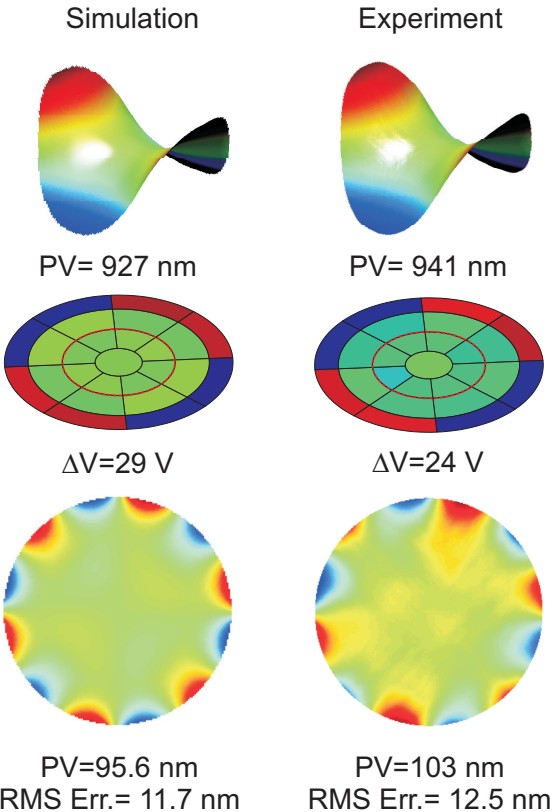

**Figure 30.** Comparison of the numerical simulation and experiment for an incremental surface of a 1 μm astigmatism. From top to bottom: surface figure, voltage map, and residual error. The result is shown in an optical pupil 34.5 mm in diameter (marked in red on the voltage map).

The second problem is associated with thermal stability: The unimorph architecture is inherently sensitive to temperature variations because of the thermal differential expansion between the PZT and the Si substrate. The ability to actively correct the effect of a temperature change $\Delta T$ is measured by the ratio between the maximum piezoelectric strain and the thermal differential expansion:

$$\frac{d_{31} E_{max}}{\Delta \alpha \, \Delta T} \tag{16}$$

Thermal stability is less critical for Earth-based telescopes because the range of operating temperature ($\Delta T$) is limited to a few degrees, but it becomes an issue for space telescopes where the operating temperature may vary much more during the orbit. Various solutions for improving thermal stability have been investigated in [24]. Figure 31 shows an ultra-flat design, which has been obtained by changing the position of the piston-tip–tilt actuators; this configuration is more compact than that of Figure 29 and has a first mode at 350 Hz.

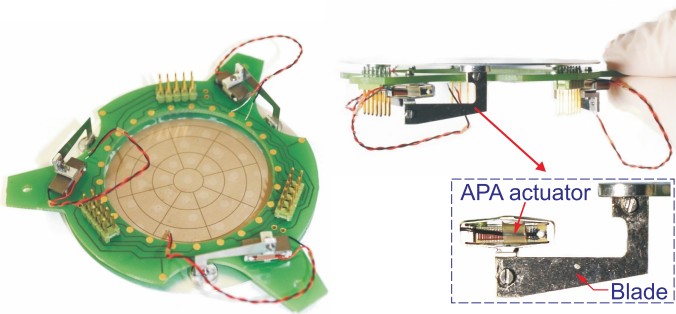

**Figure 31.** Lightweight and ultra-flat design with a first mode at 350 Hz.

## 10. Adaptive Spherical Shell

Flexible ultra-thin deployable reflectors are the subject of increasing interest because they combine a low areal density with the capability to deploy to sizes significantly larger than their stowed size, allowing a better resolution ($\sim\lambda/D$) and a better signal quality ($\sim D^2$). This includes large sizes, which would not be possible with the current technology, even with deployable segmented optics similar to that of the James Webb Space Telescope (JWST has a primary mirror of 6.5 m), but also nanosatellites and CubeSats which are becoming increasingly popular for missions once carried out by much bigger satellites (e.g., [25]). Figure 32 shows a conceptual view of a deployable reflector with flexible petals in folded (left) and deployed (right) configurations. The deployment is achieved with a translation mechanism moving the reflector outside the CubeSat; the reflector, initially constrained inside the CubeSat is freed gradually as it moves upwards and the petals return elastically to their original spherical shape. A precise spherical shape will require some active control, which is achieved with a distributed layer of piezoelectric material at the back of the reflector.

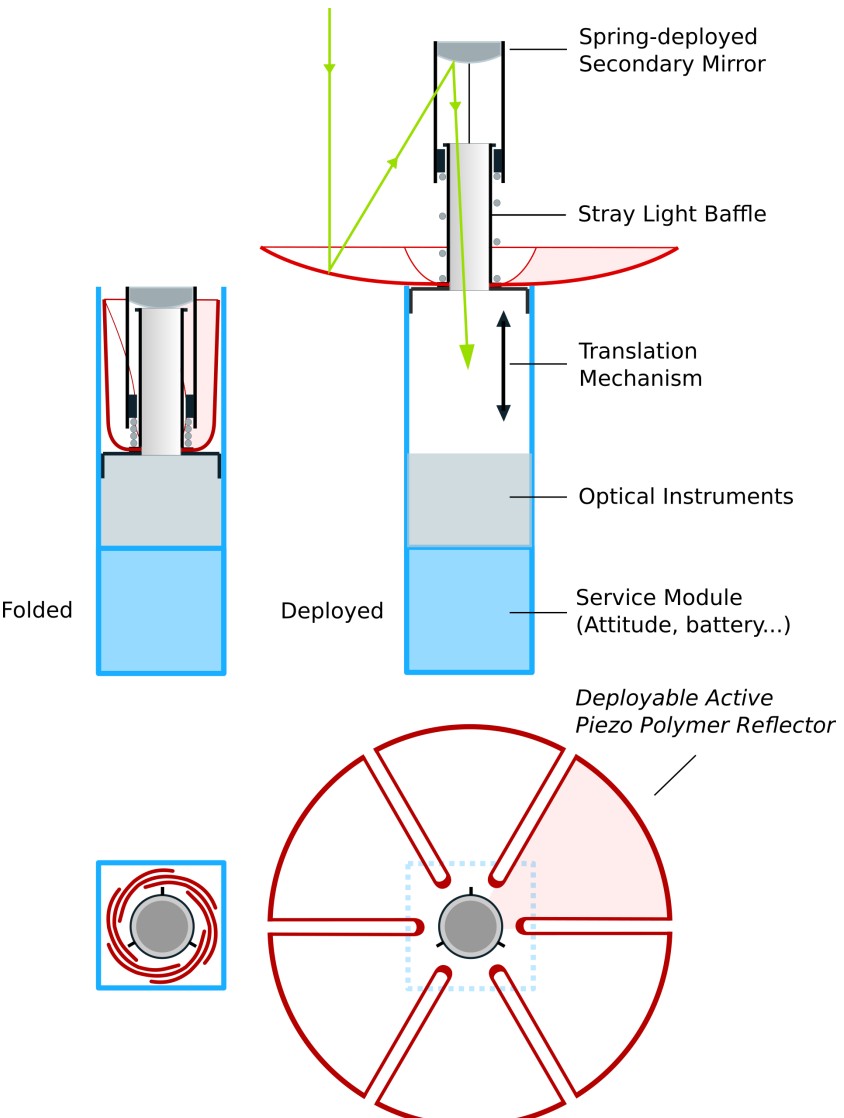

**Figure 32.** Deployable reflector made of six petals folded in a 3U CubeSat (**left**) and deployed (**right**).

### 10.1. Adaptive Plate vs. Spherical Shell

The foregoing section illustrates how unimorph piezoelectric actuation is effective for controlling the shape of a flat AO mirror. It is therefore tempting to extend the piezoelectric

control to spherical and parabolic reflectors. However, the behavior of a spherical shell is significantly different from that of a plate, as illustrated in Figure 33. Under the action of a uniform layer of piezoelectric material, a flat plate will take a spherical shape (the curvature is given by the celebrated Stoney formula). On the contrary, a spherical shell is a lot stiffer; the amplitude of the displacements is significantly reduced and takes a shape similar to that of Figure 33b: the increment of curvature is concentrated near the edge, in a boundary layer of thickness $\varrho \sim \sqrt{R_c t}$, where $R_c$ is the radius of curvature and $t$ is the thickness of the shell. Accurate shape control of a spherical shell with an array of piezoelectric actuators requires that the electrode size be smaller than the boundary layer thickness $\varrho$. Any significant departure from this condition will lead to a wavy reflector shape at the transition between electrodes excited with different voltages; thus, more curvature and a thinner shell will mean more electrodes, more complex metrology, and a control algorithm.

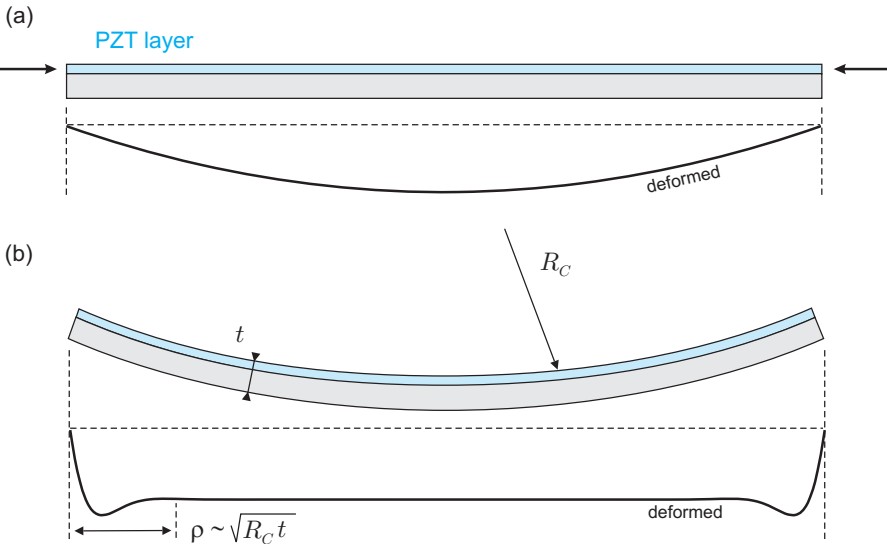

**Figure 33.** (**a**) Uniform thin film of active strain material acting on a flat circular plate: the plate deforms in a spherical shape. (**b**) Same film acting on a spherical shell: most of the additional curvature induced is concentrated near the edge.

### 10.2. Electrostrictive Polymer

The PVDF-TrFE is an electrostrictive polymer that can be spin-coated or spray-coated. After polarization, it behaves essentially as a piezoelectric material with excellent piezoelectric constant if properly annealed (up to $d_{31} = 13.54$ pC/N) [26]. In collaboration with Materia Nova, with funding from ESA, we built the two demonstrators displayed in Figure 34 where the electrode layout is also shown. The sphere has a radius of $R_c = 2.5$ m and the layer configuration is the unimorph of Figure 35a. The material properties are those of Table 3.

**Table 3.** Material properties.

| Property | PET | PVDF-TrFE | Al |
|---|---|---|---|
| Young Mod. $Y$ [GPa] | 5.6 | 2.5 | 70 |
| Poisson $\nu$ [/] | 0.38 | 0.34 | 0.32 |
| Dielectric const. $\epsilon_r$ [/] | - | 10 | - |
| Piezo const. $d_{31}$ [pC/N] | - | 15 | - |
| CTE [$10^{-6}$K$^{-1}$] | 30 | 140 | 23 |

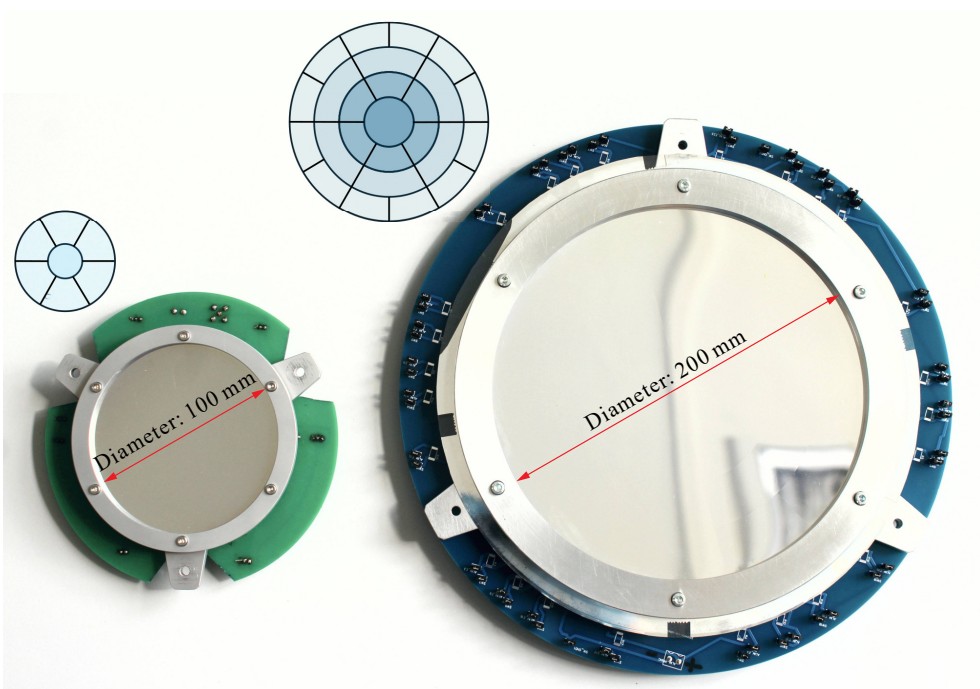

**Figure 34.** Spherical shell demonstrators built in the project MATS [27] and electrode layout.

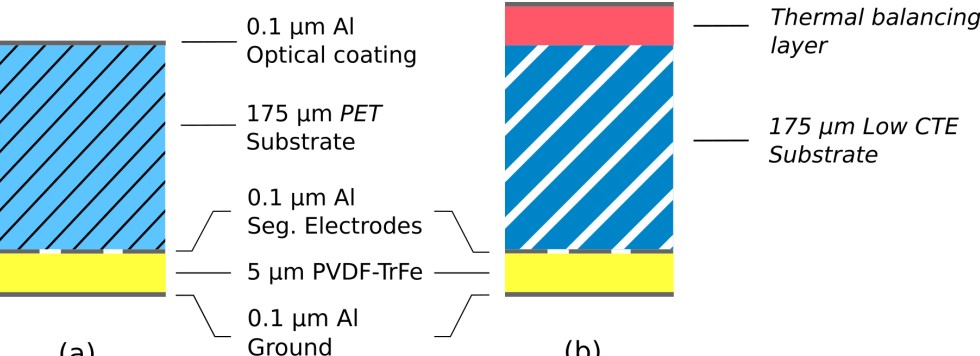

**Figure 35.** Layer configurations (**a**) unimorph used in MATS. (**b**) with thermal balancing and low CTE substrate.

## 10.3. Influence Functions, Jacobian

Figure 36 shows the experimental influence functions of six electrodes of the 200 mm demonstrator, for a voltage of 100 V. The figure also includes a comparison with a numerical simulation [27]. Once the influence functions are known, the Jacobian is easily constructed and Equation (15) can be applied to control the aberration.

## 10.4. Thermal Response

Once in orbit, the thermal disturbances are, by far, the largest contributor to the shape distortion. Looking at the last line of Table 3, one sees that the large value of the CTE of PVDF-TrFE makes the unimorph design extremely sensitive to even small temperature changes, either uniform or a gradient. This situation can be alleviated by adding a thermal balancing layer and using a low CTE substrate (Figure 35b). Numerical results reported in [28] show that this can reduce drastically the thermal sensitivity, making feasible deployable reflectors such as that shown in Figure 32. This is the topic of our current research, along with our partner Materia Nova.

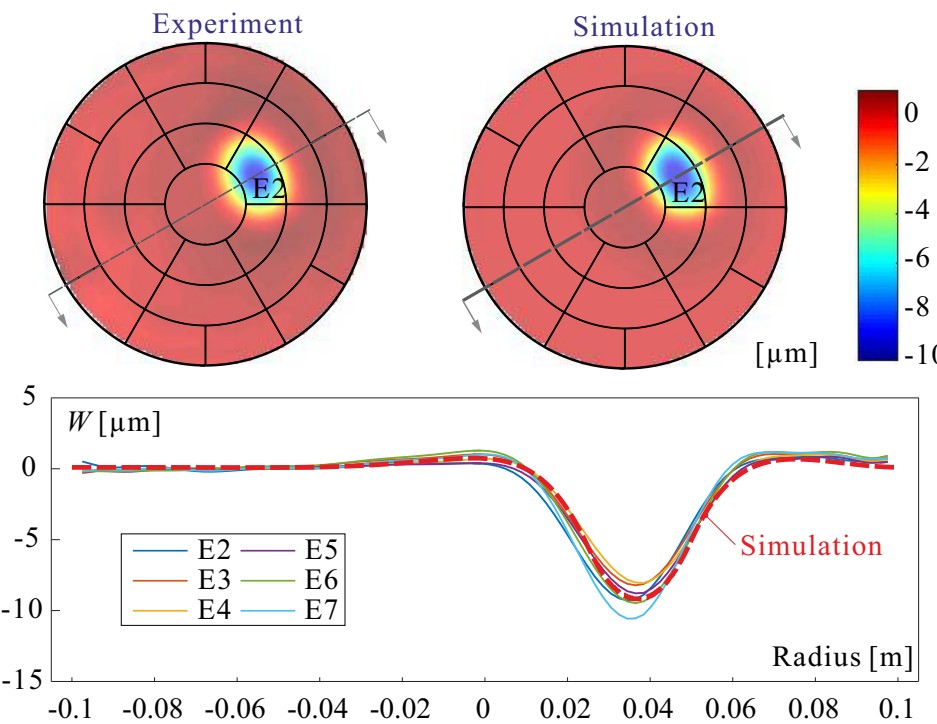

**Figure 36.** Comparison between experimental 30° cross sections of the influence functions of the six electrodes of the same row and a numerical simulation. The voltage applied is 100 V.

**Funding:** This research was funded by ESA-ESTEC GSTP program.

**Data Availability Statement:** Not applicable.

**Acknowledgments:** I wish to thank all MSc and Ph.D. students who contributed to these projects; listing them all would be impossible. I particularly enjoyed working with A. Abu-Hanieh, Y. Achkire, D. Alaluf, R. Bastaits, B. Mokrani, G. Rodrigues, and K. Wang. Visiting professors I. Burda, M. Horodinca, and I. Romanescu manufactured most of the demonstrators developed in the lab. Most of the projects reported here were supported financially by ESA in the frame of the GSTP program. The support of Belspo is deeply appreciated.

**Conflicts of Interest:** The authors declare no conflict of interest.

## Abbreviations

The following abbreviations are used in this manuscript:

| | |
|---|---|
| AO | adaptive optics |
| CFRP | carbon fiber reinforced polymer |
| FRF | frequency response function |
| IFF | integral force feedback |
| GSTP | general support technology program |
| JPL | jet propulsion laboratory |
| JWST | James Webb Space Telescope |
| LEO | low earth orbit |
| LiDAR | light detection and ranging |
| MATS | multilayer adaptive thin shell reflectors |
| MPI | micro-precision interferometer |
| PVDF-TrFE | piezoelectric polymer |
| PZT | piezoelectric ceramic |
| RMS | root mean square |
| SVD | singular value decomposition |
| ULB | Université Libre de Bruxelles |

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
