# Peer review of "Active Damping, Vibration Isolation, and Shape Control of Space Structures: A Tutorial"

_actuators, doi:10.3390/act12030122_

Round 1
Reviewer 1 Report
The paper is very well written. It illustrates the many remarkable contributions of the author to this field of research.
I only have 2 important criticisms to make:
- - The works presented are centered solely on the studies directed by Professor Preumont. As interesting as they are, they are not the only ones. And these advances are not contextualized, or compared to other state-of-the-art work. The manner of presentation leads us to believe that these are the only possible solutions. The title a “tutorial” seems a little presumptuous to me.
- - A very large part of the results presented has already been published many times. And most of them are already synthesized in the many books written by Professor Preumont. One can ask the question of the interest of such a paper for the journal?
Reviewer 2 Report
The paper presents a review of some work. I suggest to improve the manuscript by:
1) The author could sumarize the works presented in the manuscript. These works can be related and motivated. In addition, the mains contributions of these work can also be summarized in the introduction.
2) There are some practical issues that can be mentioned. For example, the author presents an integral force control to achieve damping. Sometimes, the real implementation of this control presents could present some problems. Thus, this ideal integral feedback could be substituted by a lossy integrator. In my opinion, the possible implementation issues could be included.
